# The Use of Anaplastic Lymphoma Kinase Inhibitors in Non-Small-Cell Lung Cancer Treatment—Literature Review

**DOI:** 10.3390/biomedicines12102308

**Published:** 2024-10-11

**Authors:** Anita Gorzelak-Magiera, Małgorzata Domagała-Haduch, Jacek Kabut, Iwona Gisterek-Grocholska

**Affiliations:** Department of Oncology and Radiotherapy, Medical University of Silesia, 40-615 Katowice, Poland; malgorzatadom@interia.pl (M.D.-H.); jacekkabut@gmail.com (J.K.); igisterek@sum.edu.pl (I.G.-G.)

**Keywords:** ALK-TKI, lung cancer, ALK detection, targeted therapy

## Abstract

Lung cancer is the leading cause of cancer-related morbidity and mortality. The median survival time for patients with advanced non-small-cell lung cancer before the era of molecular-based personalized treatment was 7.9 months. The discovery of predictive factors and the introduction of molecular diagnostics into daily practice made a breakthrough, enabling several years of survival in patients with advanced disease. The discovery of rearrangements in the *ALK* gene and ALK tyrosine kinase inhibitors has resulted in a dramatic improvement in the prognosis of patients with this subtype of cancer. Currently, three generations of ALK inhibitors differing in activity, toxicity and degree of penetration into the central nervous system are available in clinical practice. The current state of knowledge on ALK inhibitors used in clinical practice is summarised in this research paper. Methods of diagnosis of abnormalities in *ALK* have been shown, and the review of research that contributed to the development of the next generation of ALK inhibitors has been presented.

## 1. Introduction

Anaplastic lymphoma kinase (ALK; CD246) is a tyrosine kinase belonging to the insulin receptor family. Structurally, the native 200 kDa ALK protein is composed of an extracellular domain, a trans-membrane helix and an intracellular tyrosine kinase domain [1]. The *ALK* gene localizes within the short arm of chromosome 2. Postnatal *ALK* expression is detected physiologically at low levels only within the central nervous system. High expression in fetal life is associated with participation in the development of the central nervous system [1,2]. Activation of ALK signaling in a cancer cell can occur through gene fusion, amplification or point-activating mutation. *ALK* was first described as a fusion component detected in anaplastic large cell lymphoma (ALCL) cells [translocation (2,5)] with nucleophysin (NPM) [3]. This type of fusion accounts for 72–85% of ALK gene fusions in ALCL [1]. It is not entirely clear why *ALK* is a hot spot [“hot spot”] for translocation at so many different loci. The effect of creating an *ALK* fusion with a translocation partner is to stimulate the level of gene expression and through multimerization of the ALK kinase domain, leading to the promotion of cell differentiation and proliferation while inhibiting apoptosis (Appendix A). Since the discovery of the *ALK-NPM* fusion, the presence of more than a dozen other translocations contributing to the development of solid and hematologic cancers has been confirmed. Examples of cancers in which the presence of rearrangements within the ALK gene has been confirmed include non-small-cell lung cancer, breast cancer, colorectal cancer, kidney cancer, ovarian cancer and esophageal cancer [4]. In 2008, another mechanism promoting *ALK* gene-mediated oncogenesis in neuroblastoma cells associated with a point mutation within the tyrosine kinase was discovered [5].

## 2. Fusions of ALK in NDRP

EML4 (echinoderm microtubule-associated protein-like 4)-ALK fusion was first described in NDRP in 2007 [6]. It occurs in 3–6% of patients with adenocarcinoma lung cancer [7]. The predominant patients in this group are those with the adenocarcinoma subtype, are younger and have no or a short history of smoking [8]. Up to half of patients with the solid pattern and >10% signet-ring cells subtype may exhibit the presence of ALK rearrangement. The presence of ALK mutations is associated with a higher predilection for CNS metastasis, a higher risk of thromboembolic incidents, radiological presence of pleural effusion, central tumor location, absence of pleural tail and large pleural effusion [9,10]. The EML4-ALK fusion is estimated to account for 90% of the rearrangements within the ALK gene of NDRP [11].

The animal model for studying the EML4-ALK fusion in non-small-cell lung cancer was described by a team of researchers Soda et al. in 2007. Mouse 3T3 fibroblasts forced to express EML4-ALK fusion generated transformed foci in culture and subcutaneous tumors in nude mice [6].

Other less common rearrangements detected in lung cancer patients include KIF5B-ALK (kinesin family member 5B-ALK), TFG (trafficking from ER to golgi regulator)-ALK, KLC1 (kinesin light chain-1)-ALK, PTPN3 (protein tyrosine phosphatase non-receptor type 3)-ALK, STRN (striatin)-ALK [4].

## 3. Detection of ALK Protein or ALK Rearrangement in Patients with Non-Small-Cell Lung Cancer

An inexpensive and simple diagnostic method is immunohistochemical staining with an anti-ALK monoclonal antibody using the D5F3 cell clone. Since normal cells do not contain ALK protein, staining confirmed by light microscopy confirms the presence of abnormalities in the ALK gene. It is worth noting that anaplastic lymphoma cells, unlike lung cancer cells, show very strong ALK expression, and therefore, different antibodies are used in ALCL diagnosis than in NDRP [12,13]. FISH (fluorescent in situ hybridization) is a cytogenetic method that uses molecular probes stained with fluorochromes complementary to the ALK gene sequence. Separation of the signals from the probes by fluorescence microscopy confirms the presence of rearrangements. This provides information on whether a break has occurred within the DNA without determining the type of rearrangement. Limitations of FISH include the rapid disappearance of light signals, the need for evaluation under a fluorescence microscope, and subjective interpretation, including borderline results [12,13,14]. 

Chromogenic in situ hybridization (CISH) is also useful for ALK detection. Its efficiency is comparable in ALK rearrangement detection with the FISH method. This method allows simultaneous analysis of histological features and gene rearrangement with fewer technical requirements—assessment is performed under a light microscope [13,14]. The reverse transcription PCR method, i.e., RT-PCR, is a highly specific method, though it is rarely used due to the high rate of false-negative results and the need for careful preservation of the material to be tested due to the labile nature of the genetic material [12,14]. A retrospective analysis of material by FISH showed 33% false-negative results [12]. 

NGS, or next-generation sequencing, is an expensive and time-consuming method. It allows for the simultaneous examination of multiple genes and the detection of fusions with any partner [12]. Two techniques are used: amplicon-based and hybrid capture-based. The first is faster, technically easier, and allows the detection of mutations that occur less frequently but are less sensitive. The second is more time-consuming and more technically demanding, but it has higher sensitivity [12].

Diagnostics of lung cancer, including ALK-positive lung cancer, are still developing. New noteworthy diagnostic methods include, for example, cancer detection by a photoelectrochemical MoS2 biosensing chip [15].

## 4. ALK Inhibitors Review

### 4.1. First-Generation Inhibitors

#### Critozinib

Crizotinib is the first ALK receptor inhibitor discovered in 2005. Structurally, it is a small-molecule tyrosine kinase inhibitor with an affinity for the MET, ROS-1 and ALK receptors [16] (Appendix A). The Phase I study NCT00585195 showed an objective response rate in 61% of treated patients with a median time to disease progression (PFS) of 9.7 months [17]. Safety, efficacy and impact on improving quality of life were confirmed in the PROFILE 1005 study, with the most commonly reported side effects being visual disturbances, nausea and diarrhea and vomiting [18]. The PROFILE 1014 study confirmed a significant improvement in time to progression-free survival in the crizotinib study arm (PFS 10.9 months vs. 7 months) compared to the control arm in which a platinum derivative with pemetrexed was used [19]. Similar results were obtained in the Asian population in the PROFILE 1029 trial (PFS 11.1 vs. 6.8 months) [20]. Also, in second-line treatment, after progression on platinum-based chemotherapy, crizotinib proved more effective and safer than conventional chemotherapy with docetaxel or pemetrexed (PFS 7.7 months vs. 3.0 months). The incidence of treatment-related serious adverse events (Grade 3 and 4) was similar in the two groups (33% vs. 32%) [21]; Table 1. Crizotinib received U.S. Food and Drug Administration (FDA) registration in 2011 for the treatment of non-small-cell lung cancer (NSCLC) with ALK gene rearrangement in second and subsequent lines of treatment and was approved as a first-line treatment by the FDA in 2013 [16,22]; Table 2. Low penetration across the blood–brain barrier limits the use of crizotinib in patients with metastatic CNS lesions and makes CSF a common site of disease progression [23,24]. Another limitation to efficacy is the high rate of acquired resistance through activation of alternative pathways or acquired mutations, limiting the duration of response to treatment [16].

### 4.2. Second-Generation ALK Inhibitors

#### 4.2.1. Ceritinib

In vitro, ceritinib has 20-fold greater potency against the ALK receptor than crizotinib [37]. Its next-line efficacy and toxicity profile was determined in the ASCEND-1 and ASCEND-2 trials [26,38]; Table 1. The overall CNS response rate was 45% in the ASCEND-2 trial [26]. The FDA registered ceritinib in patients after progression on crizotinib and in case of intolerance to crizotinib treatment in 2014 [22]. The ASCEND-4 trial proved the superiority of ceritinib over the combination of cisplatin and pemetrexed in patients previously untreated for ALK-positive lung adenocarcinoma. The median progression-free time was 16.6 months for ceritinib vs. 8.1 months in the chemotherapy-treated group [32]. It received registration in first-line treatment in 2017 [22]; Table 2.

#### 4.2.2. Alectinib

Alectinib is a highly selective ALK inhibitor that shows its antitumor activity in cells with ALK rearrangement, as well as in cells with acquired resistance mutations. With a similar potential to inhibit ALK, it has an affinity for the tyrosine kinase RET [39,40] (Appendix A). Due to the fact that it is not a substrate of P-glycoprotein or breast cancer resistance protein (BCRP), it penetrates the CNS which has contributed to prolonging the time to disease progression in the CNS [41,42]. The ALUR trial showed a significant efficacy advantage of alectinib over docetaxel or pemetrexed in patients previously treated with crizotinib. The CNS response rate in the alectinib arm was 54.2% vs. 0% in the crizotinib arm, with a lower treatment toxicity rate (CTC3 ≥ 3 41.2% vs. 27.1%) [29]. In the final efficacy date published in 2022, ORR was 66.7% with alectinib arm versus 0% with chemotherapy (*p* < 0.001) [43]. Registration by the FDA was initially obtained for second-line treatment in 2015 [44]. Then, based on the results of the ALEX trial, which showed an advantage for alectinib in a 53% reduction in the risk of disease progression or death compared to crizotinib, the drug was approved by the FDA for first-line treatment two years later; Table 2. Treatment toxicities in patients taking alectinib were lower than those reported in patients treated with crizotinib [30]. In 2024, alectinib was approved by the FDA for the add-on treatment of ALK-positive patients. Registration was based on the results of the ALINA trial, in which the 2-year disease-free survival rate of patients with stage II or III disease was 93.8% for those receiving alectinib and 63% in the group treated with chemotherapy alone [45]. A real-world data (RWD) study showed higher efficacy of alectinib than ceritinib; nevertheless, in that study, patients receiving alectinib were younger, had received more lines of treatment earlier and had lower performance status compared to patients treated with ceritinib, which does not allow for drawing accurate conclusions [46].

#### 4.2.3. Brigatinib

A second-generation ALK inhibitor with an action potential 12 times more potent than crizotinib in in vitro studies (Appendix A). It shows activity independently of more than a dozen resistance mutations, including the most common ones, i.e., G1202R or L1196M [47,48].

Efficacy in a population previously treated with crizotinib was proven in the phase II ALTA trial [49]. It was registered by the FDA in 2017 for the next line of treatment. The first-line registration was based on the results of the ALTA-1L trial, in which the first-line treatment was brigatinib or crizotinib. It achieved 3-year PFS in 43% of brigatinib-treated patients and 19% in the crizotinib-treated group. In addition, brigatinib achieved significantly better results in the percentage of objective CNS responses. It showed a 56% reduction in intracranial progression and as much as a 71% reduction in patients who had already been diagnosed with CNS metastases before starting treatment. Brigatinib, compared to crizotinib, achieved its superior efficacy regardless of the rearrangement variant and TP53 mutation. High-grade irAEs (Grades 3–5) occurred in 78% of patients in the brigatinib group vs. 64% of patients in the crizotinib group [31]. The FDA approved the drug for first-line treatment in 2020; Table 2. The ALTA-3 trial comparing brigatinib with alectinib in patients previously treated with crizotinib showed no advantage for either drug (PFS 19.3 months for brigatinib vs. 19.2 months). Of note, however, the brigatinib-treated group had a significantly higher incidence of grade 3 or higher adverse events (44% vs. 18%). Interstitial pneumonitis was reported in 6% of brigatinib-treated patients [25]; Table 1.

#### 4.2.4. Ensartinib

A second-generation ALK inhibitor shows activity against ALK, ROS1 and MET. It has proven effective in patients not previously treated with ALK inhibitors as well as those previously treated. Responses were noted both peripherally and within the central nervous system [50,51]. In the phase I/II study, the overall disease progression-free time was 9.2 months. After subgrouping, the highest benefit was 26.2 months for patients not previously treated with ALK inhibitors PFS, followed by 9 months for patients previously treated with crizotinib and 1.9 months for patients treated with crizotinib followed by a second-generation ALK inhibitor [27]; Table 2. In the phase III eXalt3 study published in September 2021, the median progression-free time was 25.8 vs. 12.7 months in the crizotinib-treated arm. CNS response rates were also significantly longer (64% vs. 21%) [34].

### 4.3. Third-Generation ALK Inhibitors

#### 4.3.1. Lorlatinib

Lorlatinib is a third-generation tyrosine kinase inhibitor directed against ALK and ROS-1 (Appendix A). Structurally, it is a macrocyclic inhibitor, capable of penetrating the blood–brain barrier and effective against most mutations acquired during previous lines of anti-ALK treatment. It shows superior efficacy compared to crizotinib, regardless of the type of ALK gene rearrangement or mutation, EML4-ALK rearrangement variant or bypass resistance alterations [22,35,52]. The efficacy of lorlatinib has been demonstrated both in patients after failure on treatment with second-generation ALK inhibitors and in previously untreated patients [36,52]. Lorlatinib in patients previously treated with ALK inhibitors shows an objective response rate of 39.6% and in the CNS 56.1%. In the subgroup analysis, patients previously receiving an ALK inhibitor other than crizotinib are evaluated with ORR 42.9% intracranial ORR 66.7% and extracranial ORR 32.1% [53]. The CROWN trial, the first results of which were published in 2020, proved a significant advantage of lorlatinib over crizotinib in first-line treatment. A total of 78% of those receiving the drug had no disease progression in the first 12 months of treatment vs. 39% treated with crizotinib. In addition, 71% of those treated with lorlatinib had a complete CNS response [28]. An update of the study results was published in 2024. After 5 years of follow-up, median PFS was not reached in the lorlatinib-treated group [35]; Table 2.

#### 4.3.2. ALK Resistance

Despite the introduction of newer and newer generations of AKL TKIs, the primary cause of treatment failure in ALK + patients is cancer progression caused by the generation of various resistance mechanisms. The basic division of resistance to anti-ALK treatment divides resistance into primary and secondary resistance. Primary resistance is a condition in which no response to treatment is observed, and cancer progression occurs within the first three months of treatment [54]. The mechanisms of primary resistance are not fully understood. One possible cause of primary resistance is the presence of BIM with missing polymorphisms. BIM or BCl-2-like (B lymphocytoma-2) is a protein involved in the activation of programmed cell death. It has been shown that when patients have ALK + BIM with missing polymorphisms, median PFS is significantly shorter compared to patients with BIM protein without missing polymorphisms—83 days vs. 305 days, *p* = 0.03 [55]. Another potential cause of primary resistance in ALK + patients is amplification of the MYC gene—an experimental study showed an association between MYC amplification and resistance to anti-ALK therapy [56]. The literature also points out that certain EML4-ALK gene fusion variants may be associated with a lack of response to therapy. Such a relationship was shown by the T Yoshida study—in a group of 55 patients with different variants of the ALK gene, the best response to crizotinib treatment was achieved by patients with variant 1. In this group, the median PFS was 11 months, while in the group with variants other than 1, it was only 4.2 months (*p* < 0.05) [57]. In addition, the complexity of some mutations can result in false-positive results at the diagnostic stage of ALK + tumors and, consequently, a lack of response to treatment [58]. Secondary resistance is a condition in which an initial response to therapy is followed by loss of benefit from anti-ALK treatment. In the case of secondary resistance, a distinction is made between two mechanisms for its occurrence: ALK-dependent (“on target”) resistance and ALK-independent (“off target”) resistance [54]. In ALK-dependent resistance, cell proliferation remains dependent on ALK signaling pathways [55]. In ALK-independent resistance, tumor cell proliferation relies on alternative ALK-independent pathways. It is estimated that ALK-independent resistance accounts for 40% of cases of resistance to anti-ALK therapy [59]. Among the mechanisms of ALK-dependent resistance are structural changes in the kinase domain that block drug binding and/or alterations in ATP binding. One of the better-understood mutations is ALKG1202R, found in many cases of acquired resistance to treatment with second-generation ALK inhibitors; moreover, it has been experimentally confirmed that the occurrence of this mutation after treatment with ceritinib is a predictor of response to third-generation anti-ALK therapy in the form of lorlatinib [60]. Another mutation in the ALK gene—at codon I1171—is most likely to result in secondary resistance to the use of first- and second-generation drugs [61]. To date, a total of dozens of mutations in the ALK gene conditioning resistance to anti-ALK therapy have been described. ALK-independent resistance is conditioned by a number of distinct mechanisms [62]. Overexpression of the protein glycoprotein-P, encoded by the MDR-1 gene, is considered to be one of them. When it is amplified/induced, an excess of synthesized glycoprotein P is responsible for the removal of cytotoxic drugs such as taxoids or vinca alkaloids from cells [63]. A correlation between P-glycoprotein overexpression and ceritninib resistance was already demonstrated in 2015. Moreover, the use of P-glycoprotein inhibitors resulted in the re-sensitization of cells to ceritinib [64]. In other cases, ALK-independent resistance is caused by the generation of signaling pathways alternative to ALK. It has been shown that one mechanism of resistance to crizotinib is increased autophosphorylation of the epidermal growth factor receptor (EGFR). This leads to the activation of downstream signaling pathways despite the absence of mutations in the EGFR gene [59]. Another paper highlights the roles of the MAPK signaling pathway—its reactivation may be crucial in the formation of ALK-independent secondary resistance. In a study in mice, the combination of crizotinib with trametinib (a MEK inhibitor) resulted in an enhanced anti-tumor response [65]. A link between resistance to anti-ALK therapies has also been demonstrated for SHP2, a non-receptor tyrosine phosphatase. This protein is involved in the activation of many tyrosine kinases that promote resistance to anti-ALK therapies, including the RAS-MAPK pathway. In contrast, inhibition of the SHP2 protein simultaneously with anti-ALK therapy improves treatment response by delaying the onset of treatment resistance [66]. Potential mechanisms of treatment resistance in the literature also include histological transformation. The theoretical basis for this phenomenon is the occurrence of epithelial–mesenchymal transition (EMT), which is an important process affecting tumor progression and the ability to form metastases [67]. In ALK+ tumors, transfection potentially affects the loss of a molecular target for anti-ALK drugs. Clinical confirmation of the EMT phenomenon is provided by cases of patients described in the literature in whom the transformation of ALK+ lung cancer into other histological subtypes of lung cancer has been proven [68,69]. From a clinical point of view, a type of resistance worth noting is resistance manifested by oligoprogression. Oligoprogression is a condition in which a limited number of tumor foci (3–5) undergo progression while the rest respond to anticancer therapy [70]. The causes of oligoprogression are attributed to the different exposure of tumor cells to drugs, depending on the anatomical location of the foci and also due to the heterogeneity of the tumors. In ALK-positive cancers treated with crizotinib, progression in the central nervous system is common, which is due to the low concentration of the drug in the cerebrospinal fluid [24]. The treatment of choice for oligoprogression of ALK-positive cancer is local treatment with radiotherapy and maintenance of current systemic therapy. The efficacy of such management has been demonstrated for both cerebral oligoprogression and limited progression of extracranial lesions [71,72].

## 5. Discussion

Currently, we have three generations of ALK inhibitors available for patients with non-small-cell lung cancer. The successive generations differ from the previous ones by increasing efficacy, better penetration into the CNS, and the ability to overcome an increasing number of resistance mutations. The first of this group of drugs was crizotinib, which improved the quality and length of life of patients with NDRP and achieved a PFS of 12 months. The ALEX trial comparing the treatment efficacy of crizotinib with alectinib proved that with the ability to break resistance and improve penetration of the next generation of ALK inhibitors, it is possible to significantly increase the length of time to disease progression as well as overall survival. The undisputed therapeutic success of ALK inhibitors was crowned by the CROWN trial in which, in the group treated with lorlatinib, a 5-year progression-free survival rate was achieved by 60% of the subjects. Currently, it seems that the two most commonly chosen ALK inhibitors for first-line treatment are aceitinib and lorlatinib. The 3-year progression-free time of 64% for lorlatinib and 46.4% for alectinib, as well as the higher rate of CNS complete responses in the lorlatinib trial, support its superiority. ALK inhibitors exemplify the breakthroughs in patient prognosis that have been made over the years by personalizing the treatment of cancer patients and breaking through resistance mechanisms that contributed to early treatment failure.

## Figures and Tables

**Table 1 biomedicines-12-02308-t001:** The most common ALK-TKI-adverse events.

Inhibitor ALK	Any Grade	Grade 3 or 4
Crizotinib [21]	Vision disorder 60% Diarrhea 60% Nausea 55% Vomiting 47% Constipation 42% Elevated transaminases 38% Edema 31% Fatigue 27% Upper respiratory infection 26% Dysgeusia 26%	Elevated transaminases 16% Dyspnea 4% Constipation 2% Nausea 1% Vomiting 1%
Alectinib [25]	Increased AST 38% Increased ALT 36% Increased blood bilirubin 29% Increased blood CPK 29% Anemia 25% Peripheral edema 13% Constipation 23% Lipase increased 12% Myalgia 11% Fatigue 9%	Increased ALT 6% Increased AST 5% Anemia 2% Increased blood bilirubin 2% Increased blood CPK 2% Lipase increased 2% Increased amylase 2%
Brigatinib [25]	Increased blood CPK 70% Increased AST 53% Increased ALT 40% Hypertension 22% Lipase increased 22% Rash 10% Interstitial lung disease 6% Myalgia 5% Fatigue 4%	Increased blood CPK 36% Lipase increased 7% Hypertension 6% Increased ALT 3% Rash 1% Increased AST 1%
Ceritinib [26]	Nausea 81% Diarrhea 80% Vomiting 63% ALT increased 44% Decreased appetite 41% Fatigue 36% Weight decreased 34% AST increased 32% Abdominal pain 31% Constipation 29% Cough 21% Pyrexia 21% Dyspnea 21%	ALT increased 17% γ-GT increased 12% Nausea 6% Diarrhea 6% Fatigue 6% AST increased 5% Pyrexia 3% Dyspnea 6%
Ensartinib [27]	Rash 56% Nausea 36% Pruritus 28% Vomiting 26% Fatigue 22% Dysgeusia 19% Edema 15%	Rash 12% Pruritus 5% Fatigue 2% Vomiting 1% Nausea 1% Decreased Appetite 1% Edema 1%
Lorlatinib [28]	Hypercholesterolaemia 70% Hypertriglyceridaemia 64% Oedema 55% Weight increased 38% Peripheral neuropathy 34% Cognitive effects 21% Diarrhoea 21% Anaemia 19% Fatigue 19% Hypertension 19% Vision disorder 18% Mood affects 16%	Hypertriglyceridaemia 20% Hypercholesterolaemia 16% Weight increased 17% Hypertension 10% Oedema 4% Anaemia 3% Peripheral neuropathy 2% Cognitive effects 2% Diarrhoea 1% Fatigue 1% Mood effects 1%

**Table 2 biomedicines-12-02308-t002:** Overview of FDA-approved ALK inhibitors with research studies.

Drug	Study	Phase Trial	Study Design	Results	Line of Treatment	FDA Approval
CRIZOTINIB [19,21]	PROFILE 1014	phase III trial	crizotinib vs. chemotherapy pemetrexed + cisplatin/carboplatin	PFS 10.9 months vs. 7.0 months	First line	2013
	PROFILE 1007	phase III trial	crizotinib vs. chemotherapy pemetrexed or docetaxel	PFS 7.7 vs. 3.0 months	Later line	2011
ALEKTINIB [29,30]	ALEX	phase III trial	alectinib or crizotinib	34.8 vs. 10.9 months	First line	2017
	ALUR	phase III trial	alectinib vs. chemotherapy pemetrexed or docetaxel	PFS 9.6 vs. 1.4 months	Later line	2013
BRIGATINIB [25,31]	ALTA-1L	phase III trial	brigatinib versus crizotinib	3 years PFS rate was 43% in the brigatinib arm vs. 19% in the crizotinib arm	First line	2020
	ALTA-3	phase III trial	brigatinib versus alectinib	PFS was 19.3 months with brigatinib and 19.2 months	Later line	2017
CERITINIB [32,33]	ASCEND-4	phase III trial	ceritinib vs. chemotherapy (cisplatin/carboplatin plus pemetrexed	PFS 16.6 months vs. 8.1	First line	2017
	ASCEND-5	phase III trial	ceritinib vs. chemotherapy pemetrexed or docetaxel	PFS 5.4 vs. 1.6 months	Later line	2014
ENSARTNIB [34]	eXalt3	phase III trial	Ensartynib vs. crizotinib	PFS 25.8 vs. 12.7 months	First Line	2024
LORLATINIB [35,36]	CROWN	phase III trial	lorlatinib vs. crizotinib	PFS was not reached for lorlatynib with a median follow-up for PFS of 60.2 vs. 9.1 months	First line	2021
	NCT01970865	phase I/II trial	Lorlatynib in various subgroup	Previous crizotinib with or without chemotherapy group ORR 69.5%	Later line	2018

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
