# Peer review of "The Use of Anaplastic Lymphoma Kinase Inhibitors in Non-Small-Cell Lung Cancer Treatment—Literature Review"

_biomedicines, 2024, doi:10.3390/biomedicines12102308_

Round 1

Reviewer 1 Report

Comments and Suggestions for Authors

The paper titled "The use of ALK inhibitors in non-small cell lung cancer treatment - literature review" presents methods of diagnosis of abnormalities in ALK. The quality of this study is quite good. However, the article structure, grammar, need to be strengthened and revised. This paper should be revised for resubmission.

1.      What is the median progression-free survival (PFS) for patients treated with crizotinib compared to those treated with chemotherapy in the PROFILE 1014 and PROFILE 1029 trials?

2.      What percentage of patients treated with alectinib in the ALUR study experienced an improvement in CNS response rate, and how does this compare to chemotherapy?

3.      What is the incidence of grade 3 or higher adverse events in patients treated with brigatinib compared to crizotinib in the ALTA-1L trial?

4.      What percentage of patients in the CROWN trial showed no disease progression after 12 months of treatment with lorlatinib, compared to crizotinib?

5.      How does the overall response rate (ORR) of lorlatinib in the CNS compare to extracranial regions in patients previously treated with second-generation ALK inhibitors?

6.      What is the median PFS for patients treated with ensartinib compared to crizotinib in the phase III eXalt3 trial, and what was the CNS response rate in each group?

7.      For the literature review of appoach methods, we recommend the authors add some references, such as:

[1] Lai, C. L., et al. Lung cancer cells detection by a photoelectrochemical MoS 2 biosensing chip. Biomedical Optics Express15(2), 753-771 (2024).

Comments on the Quality of English Language

Minor editing of English language required.

Author Response

Thank you very much for your constructive comments. We have supplemented the article with the missing data.

Comments 1:  What is the median progression-free survival (PFS) for patients treated with crizotinib compared to those treated with chemotherapy in the PROFILE 1014 and PROFILE 1029 trials?

We agree with this comment and add necessary informations. 

The PROFILE 1014 study confirmed a significant improvement in time to progression-free survival in the crizotinib study arm (PFS 10.9 months vs. 7 months) compared to the control arm in which a platinum derivative with pemetrexed was used [21]. Similar results were obtained in the Asian population in the PROFILE 1029 trial (PFS 11.1 vs 6.8 months).

2.      What percentage of patients treated with alectinib in the ALUR study experienced an improvement in CNS response rate, and how does this compare to chemotherapy?

We agree with this comment and add necessary informations.

The CNS response rate in the alectinib arm was 54.2 % vs 0 % in crizotinib arm, with a lower treatment toxicity rate (CTC3≥3 41.2% vs 27.1%) [29]. In the final efficacy date published in 2022 year ORR was 66.7% with alectinib arm versus 0% with chemotherapy (P < 0.001).

3.  What is the incidence of grade 3 or higher adverse events in patients treated with brigatinib compared to crizotinib in the ALTA-1L trial?

We agree with this comment and add necessary informations.

High-grade irAEs (Grades 3–5) occurred in 78% of patients in the brigatinib group vs 64 % of patients in the crizotinib group.

4.      What percentage of patients in the CROWN trial showed no disease progression after 12 months of treatment with lorlatinib, compared to crizotinib?

We agree with this comment and add necessary informations.

78% of those receiving the drug had no disease progression in the first 12 months of treatment vs 39% treated with crizotinib. 

5. How does the overall response rate (ORR) of lorlatinib in the CNS compare to extracranial regions in patients previously treated with second-generation ALK inhibitors?

We agree with this comment and add necessary informations.

 In the subgroup analysis, patients previously receiving an ALK inhibitor other than crizotinib are evaluated ORR 42.9% intra cranial ORR 66.7% and extra cranial ORR 32.1%. 

6. What is the median PFS for patients treated with ensartinib compared to crizotinib in the phase III eXalt3 trial, and what was the CNS response rate in each group?

We agree with this comment and add necessary informations.

In the phase III eXalt3 study published in September 2021, the median progression-free time was 25.8 vs 12.7 months in the crizotinib-treated arm. CNS response rates were also significantly longer (64% vs 21%). 

7. For the literature review of appoach methods, we recommend the authors add some references, such as:

[1] Lai, C. L., et al. Lung cancer cells detection by a photoelectrochemical MoS 2 biosensing chip. Biomedical Optics Express15(2), 753-771 (2024).

Thank you for your comments. We have enriched the article with a citation regarding lung cancer diagnosis. 

Reviewer 2 Report

Comments and Suggestions for Authors

I recommend adding a general outline (figure) of the role of ALK in tumorigenesis, especially in non-small cell lung cancer.

Please add chemical structures of ALK inhibitors.

Interesting question: how were ALK inhibitors discovered for this specific cancer, are there animal models of non-small cell lung cancer, and how is the development of specific ALK inhibitor-based drugs for the treatment of this cancer going.

Are there other ALK inhibitors that have proven ineffective in the treatment of this lung cancer?

Author Response

Thank you very much for your constructive comments. We have supplemented the article with the missing data.

  1. I recommend adding a general outline (figure) of the role of ALK in tumorigenesis, especially in non-small cell lung cancer.

We agree with this comment and add necessary informations. 

2.  Please add chemical structures of ALK inhibitors.

We agree with this comment and add necessary informations. 

3. Interesting question: how were ALK inhibitors discovered for this specific cancer, are there animal models of non-small cell lung cancer, and how is the development of specific ALK inhibitor-based drugs for the treatment of this cancer going.

We agree with this comment and add necessary informations. 

The animal model for studying the EML4-ALK fusion in non-small cell lung cancer was described by a team of researchers Soda et all. in 2007. Mouse 3T3 fibroblasts forced to express EML4-ALK fusion generated transformed foci in culture and subcutaneous tumours in nude mice [8].

4. Are there other ALK inhibitors that have proven ineffective in the treatment of this lung cancer?

We did not find information about any ALK inhibitor that would be completely ineffective in the treatment of ALK-positive lung cancer, aside from cases of primary resistance to ALK inhibitors. The issue of primary and acquired resistance has been discussed in the article. 

Round 2

Reviewer 2 Report

Comments and Suggestions for Authors

I thank the authors for revision. In my opinion, the work can be accepted in this edition.